# Genetic Factors of Non-Obstructive Azoospermia: Consequences on Patients’ and Offspring Health

**DOI:** 10.3390/jcm10174009

**Published:** 2021-09-05

**Authors:** Csilla Krausz, Francesca Cioppi

**Affiliations:** Department of Experimental and Clinical Sciences “Mario Serio”, University of Florence, 50139 Florence, Italy; francesca.cioppi@unifi.it

**Keywords:** azoospermia, infertility, genetics, exome, WES, Y chromosome, cancer, NOA, genes, general health, ICSI, offspring health

## Abstract

Non-Obstructive Azoospermia (NOA) affects about 1% of men in the general population and is characterized by clinical heterogeneity implying the involvement of several different acquired and genetic factors. NOA men are at higher risk to be carriers of known genetic anomalies such as karyotype abnormalities and Y-chromosome microdeletions in respect to oligo-normozoospermic men. In recent years, a growing number of novel monogenic causes have been identified through Whole Exome Sequencing (WES). Genetic testing is useful for diagnostic and pre-TESE prognostic purposes as well as for its potential relevance for general health. Several epidemiological observations show a link between azoospermia and higher morbidity and mortality rate, suggesting a common etiology for NOA and some chronic diseases, including cancer. Since on average 50% of NOA patients has a positive TESE outcome, the identification of genetic factors in NOA patients has relevance also to the offspring’s health. Although still debated, the observed increased risk of certain neurodevelopmental disorders, as well as impaired cardiometabolic and reproductive health profile in children conceived with ICSI from NOA fathers may indicate the involvement of transmissible genetic factors. This review provides an update on the reproductive and general health consequences of known genetic factors causing NOA, including offspring’s health.

## 1. Introduction

Azoospermia (absence of spermatozoa in the ejaculate) is a relatively frequent cause of infertility occurring in about 1–2% of men in the general population. Its origin can be congenital or acquired and can be divided into: (i) hypothalamic–pituitary axis dysfunction, (ii) primary quantitative spermatogenic disturbances, and (iii) urogenital duct obstruction causing obstructive azoospermia (OA), including anatomic and genetic (e.g., *CFTR* mutation causes) [1]. While central hypogonadism is a rare etiology of Non-Obstructive Azoospermia (NOA), accounting for approximately 5% of cases, primary testicular failure is responsible for the large majority of azoospermia (>75%) [2].

NOA is a symptom which can be the consequence of different types of testicular failure such as: (i) Sertoli-Cell-Only Syndrome (SCOS), (ii) Maturation Arrest (MA) at different stages of germ cell maturation (such as Spermatogonial and Spermatocyte Arrest (SGA, SCA)), (iii) hypospermatogenesis; (iv) mixed forms. Similar to histology, follicle-stimulating hormone (FSH) and luteinizing hormone (LH) levels, testis volume, and degree of androgenization can vary among NOA men. This intrinsic clinical heterogeneity implies the involvement of several different acquired and congenital genetic factors. The known genetic factors underlying the NOA phenotype account for almost 30% of cases and include primarily chromosomal abnormalities (such as 47, XXY Klinefelter syndrome and 46, XX male), followed by Y-chromosome microdeletions and monogenic defects. Three comprehensive reviews on this topic were recently published providing a complete list of NOA-related genetic factors [3,4,5]. NOA is receiving a growing attention, not only because it is the most severe infertility phenotype but also because epidemiological observations show a link between azoospermia and a higher incidence of morbidity and lower life expectancy [6,7,8,9,10,11,12,13,14] (Table 1). 

It is worth noting that a 10-fold increased risk of hypogonadism among azoospermic men has been reported [15], which by itself can be linked to adverse health outcomes, i.e., higher risks of metabolic syndrome [16], cardiovascular disease [17], rheumatic autoimmune diseases [18] and overall mortality [16]. In addition, a significantly increased risk of developing testis cancer in infertile men has been well-documented [19,20]. In particular, men with azoospermia present a 2.9 times higher risk to develop cancer in respect to the general population [8]. 

Following the above observations, semen phenotype has been proposed as a biomarker of general health [12,13,19,21]. Since on average 50% of NOA patients will have a positive Testicular Sperm Extraction (TESE) outcome, the routine testing for known genetic anomalies has relevance not only for the carrier but also for his future child. Elucidating the genetic causes underlying azoospermia would allow improving the management of patients, identifying those azoospermic men who are unlikely to have testicular spermatozoa, those who are at higher risk for general health problems and would also have an impact on the health of their descendants (Figure 1).

This review focuses on the reproductive and general health consequences of known genetic factors causing NOA including offspring’s health.

## 2. Consequences of Chromosomal Anomalies 

### 2.1. Klinefelter Syndrome (47,XXY)

Is the most common genetic disorder causing NOA, which is characterized by the presence of an extra X chromosome. Its prevalence is 0.1–0.2% in newborn male infants, and it increases in relation to the age of diagnosis. Its frequency has been estimated as 3–4% among infertile males and 10–12% in azoospermic subjects [22,23]. The severity of the clinical phenotype of KS males may vary, and testosterone level, number of CAG repeats in the androgen receptor and/or supernumerary X chromosome could be involved in the clinical signs/symptoms of KS [24]. 

Reproductive consequences: the sex chromosome aneuploidy leads to a progressive deterioration of the testicular tissue and both the germinal epithelium and testosterone-producing Leydig cells are affected. There is a progressive deposition of ialine, which is responsible for the typical hard consistency of the testes. Azoospermia is present in about 95% of KS patients [25]. However, very rarely, non-mosaic KS patients can have spermatozoa in their ejaculate, leading to spontaneous pregnancy. The success rate for the recovery of spermatozoa through microsurgical TESE (m-TESE) in KS men is 34–44% [26]. As for other NOA patients, also in this case, the fertility status of the female partner is essential for achieving pregnancy through Intracytoplasmic Sperm Injection (ICSI). A growing number of KS patients are diagnosed during their fetal life, through pre-natal genetic diagnosis. This novel trend raises the issue about the correct management of these patients during their transition period from childhood to adulthood [23]. There are still debated questions such as the right timing for testosterone replacement therapy (for its potential interference with residual spermatogenesis) and m-TESE in young post-pubertal KS boys [27,28].

General health consequences: besides azoospermia, a wide spectrum of clinical manifestations including several comorbidities are present, i.e., metabolic syndrome, type 2 diabetes mellitus, anaemia, cardiovascular diseases (ischemic heart disease, deep vein thrombosis, lung embolism), osteopenia/osteoporosis, breast cancer, extra-gonadal germ cell tumours, non-Hodgkin lymphoma, haematological cancers and some autoimmune diseases and psychiatric disorders [23,25,29,30]. Part of the above pathological conditions are the consequence of impaired testosterone production (e.g., metabolic syndrome, osteopenia/osteoporosis), others may be due to X-linked gene dosage effect or epigenetic factors [3]. Given the complexity of this disease, patients care in dedicated multidisciplinary centres is advocated [23,31]. 

Consequences on offspring’s health: it is expected that spermatozoa from KS subjects are likely to be originated from euploid spermatogonia, i.e., the testis shows a mosaic condition where the majority of tubules contains 46,XXY spermatogonia while in a few of them spermatogonia carry a normal chromosomal asset (46,XY) [32]. Accordingly, data in the literature do not show an increased risk of having a KS child compared to infertile men with normal karyotype [32]. In fact, more than 200 healthy offspring were born worldwide from KS fathers and only a few cases of 47,XXY fetus/newborns were reported [33,34,35]. Despite the encouraging data that KS offspring seem not to be affected by the genetic disease of the father, it remains still an open question whether Preimplantation Genetic Diagnosis (PGD) or pre-natal genetic analyses should be recommended [23]. 

### 2.2. 46,XX Testicular/ovo-Testicular Disorder of Sex Development (DSD)

Also known as 46,XX male, referring to a rare, heterogeneous clinical condition with an incidence of about 1:20,000–25,000 male newborns [36,37]. The phenotype is largely dependent on the presence or absence of the master gene of male sex determination (*SRY*), mapping to the short arm of Y chromosome. 

Reproductive consequences: due to the lack of Y chromosome linked AZF regions, which are essential for physiological spermatogenesis, all patients with this genetic anomaly are azoospermic. In addition, the gonadal development may be affected. 

General health: apart from NOA, additional features characterize these patients. Testosterone levels may range from normal to low with increased FSH and LH levels leading to the progressive development of hypogonadism [37,38]. Short stature, due to the absence of growth-regulation genes on the Y chromosome, is also a relatively common finding. 

Consequences on offspring’s health: the chance to find spermatozoa in the testes of a 46,XX male with sperm harvesting methods is zero. If the couple desires to have children, sperm donation is the only viable option, or adoption. 

## 3. Consequences of Y-Chromosome Microdeletions 

The loss of specific chromosomal sequences on the long arm of the Y (Yq) is a the most frequent molecular genetic cause of NOA [39]. The so called AZoospermia Factor (AZF) regions [40,41] contain genes involved in spermatogenesis and their removal causes different reproductive phenotypes. Many AZF genes are multicopy genes and most of them are involved in post-transcriptional and post-translational control in germ cells [42]. The AZF regions are surrounded by highly homologous repeated sequences with the same direction, representing an optimal substrate for Non-Allelic Homologous Recombination (NAHR) leading to deletions. The frequency of AZF deletions in the general population is 1:4000 but in NOA patients it can be as high as 7–10% [39,43]. The most frequently affected region is the AZFc region accounting for >60% of deletions. Due to the peculiar structure of this region, with many potential NAHR substrates, partial deletions with different breakpoints may occur at a relatively high frequency [44]. Among them, the gr/gr deletion, removing half of the AZFc gene content, is considered a proven genetic risk factor for oligozoospermia [45]. 

Reproductive consequences: depending on which type of AZF regions is removed, the semen phenotype can be azoospermia or severe oligozoospermia [39]. The complete removal of the AZFa region (approximately 792 kb) causes SCOS, whereas the complete removal of the AZFb deletion (with the extension marker sY1192 absent) leads to meiotic arrest [46]. In both conditions the probability of finding testicular spermatozoa through TESE is virtually zero. The complete removal of the AZFc is associated with a highly variable phenotype, ranging from the complete absence of germ cells in the testis (SCOS) to severe oligozoospermia. The TESE success rate in these patients is around 50%, but it is highly variable in different reports.

General health: haploinsufficiency of the *SHOX* gene, located in the pseudoautosomal region PAR1 of the Y chromosome, has been reported by Jorgez and colleagues in men with AZF microdeletion and normal karyotype [47]. The authors proposed that AZF deletion carriers are at higher risk for incurring SHOX-haploinsufficiency, which is responsible for short stature and skeletal anomalies. This alarming finding was not confirmed in a subsequent large, multicentre study [48]. In accordance with this latter study, Castro and colleagues reported PAR abnormalities only in those AZF deletion carriers who presented concomitant karyotype anomalies (isochromosome Yp and/ or Y nullisomy) [49]. In addition to PAR abnormalities, 5/7 patients with terminal AZFbc deletion and abnormal karyotype presented neuropsychiatric disorders. The authors hypothesize that CNVs in the pseudoautosomal regions (PARs) and/or the removal of MSY genes (some of them are expressed also in the brain) may play a role in the observed neuropsychiatric disorders [49]. However, the association between neuropsychiatric disorders and terminal AZFbc deletions needs further confirmation especially in view of the lack of such neurodevelopmental disorders in 46,XX males [37].

Consequences on offspring’s health: complete AZFc and partial AZFa or AZFb deletions are compatible with the presence of spermatozoa in the ejaculate or in the testis, therefore these patients will obligatorily transmit the deletion to their male descendants. Recent meta-analysis reported a reduced fertilization rate, but a similar clinical pregnancy rate, miscarriage rate, live birth rate and baby boy rate to those couple where the male partner did not carry AZF deletions [50]. It is expected that the semen phenotype of the son will be either azoospermia or oligozoospermia, however the exact semen phenotype is not predictable, since the genetic background and exposure to environmental factors may modulate the phenotypic expression of AZFc deletions. Some studies reported an association between Yq microdeletions and an overall Y-chromosomal instability, which might result in the formation of 45,X0 bearing spermatozoa [51,52]. This finding is in accordance with the relatively high incidence of AZF deletion in patients bearing a mosaic 46,XY/45,X0 karyotype with sexual ambiguity and/or Turner stigmata [53,54,55,56]. The PGD has been performed by two groups with conflicting data about the risk of monosomy X in embryos [57,58]. The limited data on children born from AZF deletion carriers show that they are apparently healthy [59].

## 4. Consequences of Monogenic Defects

Known monogenic anomalies with definitive clinical evidence are relatively rare in NOA [3]. Among them two *X-linked* genes reached diagnostic relevance: the *AR* and the *TEX11* genes.

### 4.1. AR Gene

The androgen receptor (AR) is a DNA-binding transcription factor, which is critical for several biological functions including male sex development. Upon binding of testosterone to the cytoplasmic AR, the complex translocates into the nucleus and binds to the regulatory regions of specific chromosomal DNA sequences to activate androgen dependent genes. Mutations in *AR* gene are responsible for the androgen insensivity syndrome (AIS), with an estimated prevalence of 1:20,000 to 1:64,000 live male births [60]. This condition is associated with a high variety of phenotypes, ranging from complete androgen insensitivity (CAIS) with a female phenotype (Morris syndrome) to milder degrees of undervirilization (partial form or PAIS; Refenstein syndrome) or men with only infertility (mild form or MAIS) [61]. Beside pathogenic mutations in the coding exons of the *AR* causing AIS, a polymorphic CAG repeat in exon 1 has a functional effect on the receptor’s activity. The number of the CAG repeats is inversely associated with the ligand-induced transactivational activity of the receptor and, in physiological conditions, (CAG)n directly correlates with serum testosterone levels [62]. This polymorphism has been associated with various androgen-dependent conditions including impaired sperm production (for review see [63]). 

Reproductive consequences: in the PAIS/MAIS form of disease, patients may present with quantitative spermatogenic disturbances, i.e., azoospermia or oligozoospermia. The negative effect of longer (CAG)n on spermatogenesis is a debated issue. Although the majority of studies report a higher than average (CAG)n in infertile patients, it is not possible to define a cut-off value above which infertility risk is increased and to estimate the effect size of such a risk [63].

General health: a positive correlation between CAG repeat number and depressed mood, anxiety, and low bone mineral density with accelerated age-dependent bone loss have been reported [64,65]. Smaller CAG repeat number is associated with benign prostatic hypertrophy [66] and faster prostate growth during testosterone treatment [67]. The polymorphic range in the general population is up to 39 CAG repeats, the expansion over 39 CAG is a pathological condition leading to the Kennedy disease [68]. Kennedy disease is a rare form of X-linked spinal and bulbar muscular atrophy (SBMA), characterized by progressive neuromuscular atrophy and ataxia [69] and a progressive set up of mild androgen insensitivity associated to varying traits of hypogonadism, including gynecomastia, testicular atrophy, disorders of spermatogenesis, elevated serum gonadotropins, and diabetes mellitus [70].

Consequences on offspring’s health: *AR* mutations compatible with sperm production will be obligatory transmitted to the female offspring with potential health consequences on her future male children. Concerning the (CAG)n repeats, it is worth noting that repeat expansions are inherently dynamic, often changing size when transmitted to the next generation [71]. This phenomenon, known as clinical anticipation, explains the tendency for disease severity to increase in successive generations of a family. Patients affected by Kennedy’s disease may conceive their own biological children and, similarly to *AR* mutations, the expanded CAG repeats will be transmitted to the female child, who can generate a male offspring affected by Kennedy disease. As far as the polymorphic range of CAG repeats (up to 39 CAG) is concerned, the proposed relationship between longer CAG tract and male infertility indicates a theoretical higher risk for oligozoospermic men to conceive a female child presenting a pathological expansion of CAG repeats leading to a future son with Kennedy disease [60,71]. 

### 4.2. TEX11 Gene

This gene belongs to the family of Testis Expressed genes, and it is crucial for chromosome synapsis and formation of crossovers during meiosis. By using high-resolution array-Comparative Genomic Hybridization (a-CGH) to screen men with NOA, a recurring deletion of three exons of *TEX11* in two patients has been identified [72]. Furthermore, by sequencing *TEX11* in larger groups of azoospermic men, more disease-causing mutations were detected [72,73,74,75]. Overall, mutations in *TEX11* were identified in more than 1% of azoospermic men and in as many as 15% of patients with meiotic arrest. 

Reproductive consequences: recessive mutations in this gene lead to NOA due to MA [72,73,74]. Very recently, Krausz and colleagues demonstrated that defects in the human gene showed a complete metaphase arrest, suggested by a residual spermatocytic development together with the dramatic increase in the number of apoptotic metaphases [75].

General health: apart from NOA, no additional features have been reported in mutated men. 

Consequences on offspring’s health: the chance to find mature spermatozoa in the testes of a man carrying loss of function *TEX11* mutations is virtually zero. If the couple desires to have children, sperm donation is the only viable option, or adoption. 

### 4.3. Shared Genes between Spermatogenesis and Tumorigenesis

As stated in the introduction, an increased risk of various cancers has been documented in NOA patients which in part may be due to defects in biological pathways regulating genomic integrity [8,21,76,77,78,79,80]. It is plausible that spermatogenesis and tumorigenesis may share common genetic factors, especially those involved in stem cell renewal/differentiation, mismatch repair mechanisms and apoptosis. Particularly, germline alterations in DNA repair genes, which are fundamental for maintaining the genomic integrity and stability in the early stages of the male germline, may confer hereditable predisposition to impaired spermatogenesis and cancer. 

Recent studies integrating omics and literature search revealed a significant genetic overlap between male infertility and particular types of cancer, including urologic neoplasms/carcinomas and B cell lymphoma [81,82]. By using mouse model data such as Mouse Genome Informatics (MGI) database, the integration of human orthologues to mouse male factor infertility with a curated list of known cancer genes (COSMIC genes) has identified 25 candidate genes that may confer risk of experiencing both conditions in humans [21]. In particular, there is a five-fold enrichment of COSMIC genes in the MGI male infertility list compared with genes that are not on the MGI list, suggesting that this overlap is highly non-random [21].

Apart from the bioinformatics models and epidemiological observations, there is a growing number of genes predisposing to cancer, which have been found mutated in men affected by NOA. 

#### 4.3.1. Rare Pathogenic Mutations

A recent example is related to ***FANCA*** mutations, which may cause both the classic early onset and the rarely observed late-onset Fanconi Anaemia (FA). Both manifestations are characterized by genomic instability leading to progressive bone marrow failure, congenital malformations and predisposition to typical cancers such as head and neck squamous cell carcinoma and leukaemia [83]. By performing exome analysis in NOA patients, Krausz and colleagues (2019), identified three subjects affected by SCOS with biallelic *FANCA* mutations [79]. All three subjects were unaware about having Fanconi anaemia, although two of them showed slightly abnormal blood cell count at the time of the genetic diagnosis. This study was the first in the literature reporting the accidental finding of Late onset FA (occult FA) in the absence of severe comorbidities of FA. In fact, occult FA is usually diagnosed in subjects following the diagnosis of typical malignancies. The three patients are now under surveillance by oncohematologists. This paper showed the importance of checking blood count, especially in patients presenting idiopathic SCOS, since the combined phenotype of SCOS with borderline low blood cell count indicates a higher risk for occult FA. Given that the carrier frequency of *FANCA* defects is relatively rare in the general population, pre-ICSI screening in the female partners of male carriers is not recommended. However, in case of consanguinity in the couple PGD should be offered given the severity of FA. 

Fanconi anaemia and related malignancies can also be caused by recessive mutations in the ***XRCC2*** gene [84]. Interestingly, a homozygous *XRCC2* mutation has been reported in a consanguineous family causing isolated meiotic arrest without cancer predisposition [85]. This observation leads the authors to conclude that meiosis-specific mutations may exist when the linker region of XRCC2, essential for protein–protein interactions, is affected [85,86]. In support of this, knock-in mice carrying the same *XRCC2* mutation exhibited only meiotic arrest, leading to azoospermia in males and premature ovarian failure in females [85].

Another member of the FA pathway, ***FANCM***, involved in DNA double-strand breaks (DSB) repair, was reported as the cause of NOA [78,80]. The *FANCM* gene is significantly associated with hereditary breast and ovarian cancers [87], in line with published data on female homozygous knock-out (KO) mice [88,89]. Recessive mutations in this gene seem to cause a wide spectrum of seminal phenotypes, ranging from oligoasthenozoospermia to azoospermia due to SCOS [78,80]. 

Biallelic mutations in two other DNA DBS repair genes, ***MCM8*** and ***TEX15*** were reported in azoospermia and oligo/crypto/azoospermia, respectively [90,91,92,93]. Very recently, germline mutations in the *MCM8* gene following a recessive pattern of inheritance, were detected in cancer patients [94]. One male patient affected by Lynch syndrome with fertility problems and two patients affected by breast cancer were found to be carriers of biallelic *MCM8* mutations, suggesting a role of this gene in the germline predisposition to breast cancer and hereditary colorectal cancer (CRC) [94]. Concerning *TEX15*, a rare heterozygous mutation predicted as deleterious by four bioinformatics tools was found to be significantly associated with prostate cancer risk [95]. 

Also the *X-linked **WNK3*** gene, involved in cell signalling, survival and proliferation has been linked both to NOA and cancer [96]. *AWNK3* mutation has been found to co-segregate with NOA due to SCOS in a family from Oman [97]. Concerning the role of this gene in oncology, several *WNK3* mutations in patient-derived xenografts of colorectal cancer liver metastasis were predicted to be deleterious, which might contribute to the initiation and progression of distant metastasis [98]. 

#### 4.3.2. Genetic Polymorphisms

Besides rare mutations, common polymorphisms have been reported in a total of 8 mismatch repair genes, which could account for a shared aetiology between tumorigenesis and quantitative spermatogenic failure [21]. 

Homozygous or compound heterozygous mutations in the ***MLH1*** gene have been reported in the early-onset hereditary cancer disorder Lynch syndrome, as well as in haematological malignancies and brain tumours [99], often associated with features of neurofibromatosis type 1 (NF1) syndrome [100]. Besides its known carcinogenic role, an intronic SNP in *MLH1* seem to be a risk factor for the development of azoospermia or oligozoospermia [101].

Germline ***MLH3*** variants have been reported in hereditary Lynch syndrome-associated brain tumours patients [102], and a common polymorphism (C2531T) in the 3’UTR of the gene has been associated with clinical outcomes of colorectal cancer, in terms of increased risks of relapse or metastasis in patients with heterozygous genotype [103]. Interestingly, Xu and colleagues have observed an increased risk of azoospermia or severe oligozoospermia associated with the above-mentioned polymorphism in 3′UTR of the *MLH3* gene [104]. 

***MSH5*** has been reported as a pleiotropic susceptibility locus for lung, prostate, colorectal and serous ovarian cancers [105,106], and several polymorphisms in this gene have been associated with quantitative spermatogenic defects [101,104]. Further, one low-frequency *MSH5* variant associated with an increased risk of NOA has been reported in Han Chinese men [107]. 

Biallelic germline mutations of the ***PMS2*** gene cause the constitutional mismatch repair deficiency, characterized by early-onset malignancies [108]. In addition, a founder heterozygous frameshift mutation in the same gene is responsible for the Lynch syndrome [109]. Concerning the role of *PMS2* gene in spermatogenesis, the presence of a common polymorphism in the gene leads to a reduced interaction of MLH1 and PMS2 proteins, which may result in impaired sperm production [101].

Carriers of mutations in the ***ATM*** gene have been reported to have a higher mortality rate and an earlier age at death from cancer and ischemic heart disease than non-carriers [110]. Besides this finding, germline loss-of-function *ATM* mutations seem to be enriched in men with prostate cancer and multiple primary malignancies [111]. Concerning the role of this gene in spermatogenesis, both the homozygous and heterozygous genotypes for a common variant in the *ATM* gene promoter were associated with an increased risk for idiopathic NOA [112]. 

Two SNPs in the ***XRCC1*** gene were associated with increased bladder cancer risk among Asians [113], whilst another one, the R339Q, has been implicated in susceptibility for both idiopathic azoospermia and different types of cancer, such as hepatocellular cancer in Asians and breast cancer in Indians [114,115,116,117,118].

An identical SNP (C8092A) in 3′UTR of the ***ERCC1*** gene has independently been linked to both idiopathic azoospermia and various types of cancer, including breast carcinoma, head and neck carcinoma, adult glioma [119,120,121,122]. 

In this context, the identification of shared genetic aetiologies between azoospermia and cancer may have a significant clinical impact, for improving patient care and genetic counselling. 

## 5. Health Issues in ICSI Offspring from NOA Fathers 

The introduction of ICSI among Assisted Reproductive Techniques (ART) has opened an unforeseen perspective for fatherhood in NOA patients. NOA men may father their own biological child by using non-ejaculated spermatozoa, retrieved by conventional or micro-TESE with an average success rate of 50%. As stated above, it is well known that NOA patients are at higher risk for genetic anomalies than he general population; therefore, concerns were raised regarding offspring’s health. 

Various parameters have been evaluated in ICSI children (from birth to young adulthood) born to fathers affected by spermatogenic disturbances. 

Many reports describe a high frequency of chromosomal abnormalities in ICSI babies, especially of the sex chromosomes, even when peripheral chromosome studies in the parents are normal [123,124,125]. A possible explanation for this phenomenon could depend on the testicular tubular alteration, which may determine abnormalities in the meiotic process leading to chromosomal anomalies in the spermatozoa [126]. Therefore, other forms of chromosome diploidy beyond sex chromosomes should be expected as well [127,128]. Overall, the risk of having chromosomal abnormalities, particularly sexual chromosome aneuploidy, is approximately 1% in children conceived through ICSI, which is higher than that of naturally conceived children (~0.2%) and of those conceived with conventional in vitro fertilization (IVF) (~0.7%) (see reference in [129]). In addition, children conceived by IVF and/or ICSI are at significantly increased risk for birth defects, although no risk difference between children conceived with the two ARTs has been observed [130]. A systematic review and meta-analysis showed that congenital malformations in ICSI-conceived children when compared to naturally conceived children translates into an increased risk of 7.1% of having a malformation for individuals born after ICSI versus 4.0% for naturally conceived children [131]. The most commonly observed congenital malformation involves the genitourinary tract which is significantly more frequent in ICSI children compared to both naturally conceived children and IVF children [132,133].

Besides chromosomal and birth defects, cognitive and neurodevelopmental disorders in offspring from an ICSI father have also been evaluated [134,135]. In one study a modestly increased risk of mental retardation and autism was reported in ICSI derived children [136], but this finding was not replicated in independent studies [137,138,139]. The largest of these studies, involving 10,718 children conceived with ICSI, 19,445 children conceived with IVF and 2,510,166 spontaneously conceived children, observed the greatest risk of mental retardation in children conceived through ICSI (RR 2.35, 95%; CI = 1.03–2.09) [136]. Importantly, treatment factors, i.e., ICSI and embryo cryopreservation, also appear to influence this risk [136]. In addition, an increased risk of autism in children conceived with ICSI using surgically extracted sperm (RR 4.60, 95%; CI = 2.14–9.88) was also observed [136]. This finding was not confirmed by Kissin and colleagues in the group of children conceived with ICSI for male factor infertility (HR 1.23, 95%; CI = 0.92–1.64) [139]. On the other hand, the severity of male factor does not seem to influence the cognitive development in early childhood [140,141,142].

In addition to neurodevelopmental aspects, other long-term outcomes of children conceived via ICSI due to severe male factor have been evaluated, but findings are conflicting and it is difficult to evaluate the impact of NOA on these disorders [135]. Among the large population registry studies that have examined growth and cardiometabolic factors, there is evidence that ICSI adolescents may be at risk of increased adiposity, especially girls [143,144,145,146,147]. Very recently, in male ICSI adolescents significant higher estradiol and lower testosterone/estradiol ratio, as well as a tendency towards lower inhibin B levels, was found [148]. Concerning reproductive outcomes in men conceived with ICSI, there is some evidence for impaired spermatogenesis [149,150,151]. In fact, a Belgian study, evaluating young men in the age interval 18–22 years, found reduced semen parameters among men conceived with ICSI, reporting a median sperm count and total motile sperm count being half that of their spontaneously conceived peers [151]. In addition, ICSI men showed a tendency to have lower inhibin B levels and higher FSH levels compared with spontaneously conceived peers [151].

Despite the growing number of studies, several uncertainties remain about whether any increases in risk are due to NOA or to the ICSI procedure itself [135]. To date, the global number of babies born as a result of ART techniques, such as ICSI, is more than 8 million (ESHRE: https://www.eshre.eu/ 31 August 2021), therefore it should be of paramount importance to reach to a final conclusion on safety issues. It is expected that with the extensive use of ICSI for non-male factor, a comparison of short and long-term outcomes between ICSI children derived from male factor versus non-male factor will elucidate the impact of azoospermia on the descendant’s health. 

## 6. Conclusions

Azoospermia, the most severe form of infertility, may represent a biomarker of overall health, serving as a harbinger for higher morbidity and mortality. As reported above, certain chromosomal anomalies and gene defects underlying azoospermia can be responsible for a wide spectrum of health issues beside azoospermia, including metabolic/cardiovascular disorders, autoimmune diseases, hypogonadism, syndromic conditions and cancers. After the exclusion of all known acquired causes and after performing routine genetic testing, the etiology remains unknown in a substantial proportion of patients and it could be related to yet unidentified genetic/epigenetic factors [3]). The clinical impact of discovering such “hidden” genetic factors is important to predict not only the fertility status but also the general health of these men. For instance, by performing a-CGH analyses, a “CNV burden” (especially deletions) in idiopathic infertile patients have been reported by three research groups [152,153,154], suggesting a higher genomic instability potentially relevant also for general health. CNV burden together with the above listed shared monogenic factors could be one of the many possible explanations for the higher morbidity and lower life expectancy observed in infertile men in respect to fertile men [6,7,19,152]. Similarly to monogenic disorders, the inheritance of an unstable genome may also have clinical consequences on the offspring’s health.

Thanks to the diffusion of Whole Exome Sequencing (WES) in the frame of fruitful international collaborations, the number of genes involved in NOA is rapidly increasing [3,5,155]. Exome analysis has proven to be very efficient in diagnosing the cause of meiotic arrest [75], with potential implications for TESE prognosis. WES allowed the identification of many novel genes, potentially relevant also for tumorigenesis. It can be hypothesized that inherited genetic/epigenetic factors are responsible for the increased risk of certain neurodevelopmental disorders, as well as impaired cardiometabolic and reproductive health profile in children conceived with ICSI from NOA fathers. In this context, the discovery of genetic cause underlying azoospermia would allow not only to improve the management of NOA patients, but also to predict the clinical consequences on the offspring inheriting the certain gene defect(s) (Figure 1).

While the list of genetic defects with potential impact on general health increases, it is important to note that apart from a few exceptions, we are still missing a direct evidence for a clear-cut genetic link between NOA and higher morbidity, especially in terms of cancer predisposition. Multicentre efforts are needed in order to collect long-term follow-up data on large groups of genetically well-characterized NOA patients. Apart from the routine karyotype and Y chromosome deletion analysis, we hope that WES analysis will become soon part of the genetic diagnostic work-up of NOA patients allowing diagnosis, TESE prognosis and prevention for general health.

## Figures and Tables

**Figure 1 jcm-10-04009-f001:**
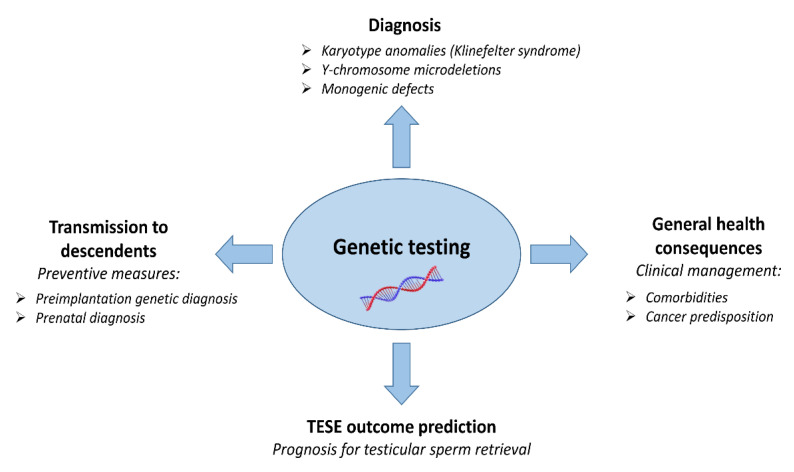
Clinical relevance of genetic testing in azoospermic men.

**Table 1 jcm-10-04009-t001:** List of studies reporting increased mortality and/or morbidity in azoospermic men.

Increased Mortality Rate (HR)	Increased Morbidity Rate (Yes/No)	Reference
n.a.	Yes *	[8]
2.29, 95% CI: 1.12–4.65	n.a.	[9]
n.a.	Yes **	[11]
3.66, 95% CI: 2.18–6.16	n.a.	[13]
2.01, 95% CI: 1.60–2.53	n.a.	[14]

HR: Hazard Ratio; n.a.: not available; * Cancer risk (HR = 2.9, 95% CI:1.4–5.4); ** The top three related-conditions are: (i) renal disease (HR = 2.26, 95% CI:1.20–4.27), (ii) alcohol abuse (HR = 1.94, 95% CI:1.11–3.39), (iii) depression (HR = 1.45, 95% CI:1.13–1.85).

## Data Availability

Not applicable.

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
