# Peer review of "Genetic Factors of Non-Obstructive Azoospermia: Consequences on Patients’ and Offspring Health"

_jcm, 2021, doi:10.3390/jcm10174009_

Round 1

Reviewer 1 Report

The goal of this paper is to evaluate the health consequences on both patients and offspring carrying various genetic defects found to be associated with nonobstructive azoospermia (NOA).  The authors provide a nice introduction to the genetic conditions and their association.

This study intends to “[focus] on the reproductive and general health consequences of known genetic factors causing NOA including offspring’s health” (page 2, line 77).  On this, the paper delivers, as the information is presented in a way in which these components are divided and easy to digest.  Most of this information is well known, and the novelty of this study suffers in the process.

After reviewing the common conditions, a number of different genetic mutations are addressed.  Most of the studies including these conditions include less than 10 patients for reference and are single studies.  It appears a larger review of each mutation should be completed to better outline, which are truly associated.  The goal is to link those traits associated with malignancy, and due to rarity it seems limited data can be presented.  If there were additional patients, this would be beneficial.

In the conclusions, the authors reference epigenetic conditions attributing for the remaining 70% of NOA cases as a point of future study.  This would certainly be a point of future investigation.  This paper serves primarily as an organized review of known conditions followed by a separate review of rare mutations in the format of case series in which NOA was also identified.

Author Response

The goal of this paper is to evaluate the health consequences on both patients and offspring carrying various genetic defects found to be associated with nonobstructive azoospermia (NOA).  The authors provide a nice introduction to the genetic conditions and their association.

This study intends to “[focus] on the reproductive and general health consequences of known genetic factors causing NOA including offspring’s health” (page 2, line 77).  On this, the paper delivers, as the information is presented in a way in which these components are divided and easy to digest.  Most of this information is well known, and the novelty of this study suffers in the process.

Yes, indeed, in the first part of the paper we deal with known genetic factors causing NOA such as chromosomal abnormalities, Y-chromosome microdeletions and the very few monogenic defects with diagnostic value. The novelty of this paper lies in providing and up-to date information on the association between the afore-mentioned genetic defects and general patient’s and offspring’s health. Since general health is the “file rouge” in this paper, in the second part we added a novel genetic perspective to the epidemiological observations reporting a higher cancer predisposition in NOA men. 

After reviewing the common conditions, a number of different genetic mutations are addressed.  Most of the studies including these conditions include less than 10 patients for reference and are single studies.  It appears a larger review of each mutation should be completed to better outline, which are truly associated.  The goal is to link those traits associated with malignancy, and due to rarity it seems limited data can be presented.  If there were additional patients, this would be beneficial.

In the second part of the study we are not dealing with diagnostic genetic factors, but we aim at providing examples from the literature supporting a possible genetic link between NOA and tumorigenesis. Of course, these are few cases at the moment, but they represent the first hints for shared genetic factors. We are confident that, with the diffusion of NGS, there will be more and more reports. We have reported all available articles published until July 2021. A part from FANCA mutations, no gene mutation(s) causing NOA was found to be responsible for the cancer onset in the same patient. In the conclusions we advocate that long term follow, especially for patients carrying mutations in cancer predispostion genes, should be performed.

In the conclusions, the authors reference epigenetic conditions attributing for the remaining 70% of NOA cases as a point of future study.  This would certainly be a point of future investigation.  This paper serves primarily as an organized review of known conditions followed by a separate review of rare mutations in the format of case series in which NOA was also identified.

We agree with the referee suggestion. This paper serves primarily as an organized review of health consequences of known genetic factors causing NOA. A future review could focus on those rare mutations causing several phenotypes including NOA.

Reviewer 2 Report

The review from Krausz and Cioppi deals with the health consequences of genetic factors associated with non-obstructive azoospermia. It is an interesting and very well-written review, but I have several suggestions which I think could improve the review.

Title: I think it would be more correct to state “Genetic factors associated with…” than “Genetic factors of…”

In the abstract I missed a statement/sentence outlining what this review deals with. E.g., “Here we review the current evidence on…” or “In this review we…”

Line 29: I know that these numbers are not exact but since this also includes obstructive cases reporting “1-2%” or “2%” could be justified here.

In the introduction, it could maybe also be relevant to just very briefly mention the CFTR gene associated with CAVD and OA.

Line 44: “Two/3” please correct.

Line 89-91: Re. prevalence of KS. Please mention that increase in prevalence by age is related to the age of diagnosis and not related to an increase in the disease by age.

Line 91-93: The severity may vary also due to other factors than the ones mentioned, and I suggest modifying the sentence to better reflect this. To my knowledge, none of the mentioned parameters can, on their own, be used to predict the severity of the KS phenotype. It is (to my knowledge) yet unknown which factors determines the severity of the KS phenotype.

Line 99: Something seems wrong with the sentence ending with “reported”. Please rephrase.

Line 118: I disagree that this is commonly accepted. Hence please reference this statement.

Line 135-136: Please also include here that the gonadal development may be affected.

In general, I find it quite intuitive to have each genetic element associated with NOA to be listed with reproductive consequences, general health, and consequences on offspring’s health. However, I miss a clearer distinction between what is likely to be caused by the testicular phenotype and what might be caused by a phenotype not related to NOA.

Line 206: I find it odd that the authors only touch briefly on other genes that have been proven causal of NOA. For a gene to reach “diagnostic relevance” is quite a somewhat subjective judgement and I suggest that the authors at least mention more genes and pathways potentially involved in NOA. Several recent publications/genes like PMID: 33963445, 34055789, 34347949, and 33980954 could be mentioned.

Author Response

The review from Krausz and Cioppi deals with the health consequences of genetic factors associated with non-obstructive azoospermia. It is an interesting and very well-written review, but I have several suggestions which I think could improve the review. 

Title: I think it would be more correct to state “Genetic factors associated with…” than “Genetic factors of…”

It was difficult to decide on the title because in the first part of this review, we have only included clear-cut genetic factors causing NOA and here we are talking about general health and offsping’s health. This is why we would like to keep the original title. 

In the second part we deal also with genes which did not reach definitive clinical evidence but the focus here is different as they are involved in cancer predisposition (see comments to the first referee). In fact, the title is – “Shared genes between spermatogenesis and tumorigenesis”.

In the abstract I missed a statement/sentence outlining what this review deals with. E.g., “Here we review the current evidence on…” or “In this review we…”

We thank the referee for the suggestion. In the abstract, we added the following sentence: “This review provides an update on the reproductive and general health consequences of known genetic factors causing NOA, including offspring’s health”.

Line 29: I know that these numbers are not exact but since this also includes obstructive cases reporting “1-2%” or “2%” could be justified here.

We agree with the referee that there are no exact numbers, we are writing about NOA when we refer to about 1%, therefore obstructive cases are not included.  Given this inexact nature of this percentage, we added in the abstract as well “about 1%”.

In the introduction, it could maybe also be relevant to just very briefly mention the CFTR gene associated with CAVD and OA.

Since the review is dealing with “genetic factors of non-obstructive azoospermia: consequences on patients' and offspring health”, we have not addressed the CFTR gene mutations associated with OA. However, in the introduction, we have now briefly mentioned the CFTR gene referring to a review considering all the causes of azoospermia.

Line 44: “Two/3” please correct.

We thank the referee for the observation. We have replaced “two/3” with “three”. 

Line 89-91: Re. prevalence of KS. Please mention that increase in prevalence by age is related to the age of diagnosis and not related to an increase in the disease by age.

We thank the referee for the comment. We have now changed the sentence as following: “Its prevalence is 0,1–0,2% in newborn male infants, and it increases in relation to the age of diagnosisIts frequency has been estimated as 3–4% among infertile males and 10–12% in azoospermic subjects [22,23]”.

Line 91-93: The severity may vary also due to other factors than the ones mentioned, and I suggest modifying the sentence to better reflect this. To my knowledge, none of the mentioned parameters can, on their own, be used to predict the severity of the KS phenotype. It is (to my knowledge) yet unknown which factors determines the severity of the KS phenotype.

According to the referee comment, we have now changed the sentence as following: “The severity of the clinical phenotype of KS males may vary, and testosterone level, number of CAG repeats in the androgen receptor and/or supernumerary X chromosome could be involved in the clinical signs/symptoms of KS.” 

Line 99: Something seems wrong with the sentence ending with “reported”. Please rephrase.

We thank the referee for the correction. We have now rephrased the sentence.  

Line 118: I disagree that this is commonly accepted. Hence please reference this statement.

We have rephrased the sentence as following: “It is expected that spermatozoa from KS subjects are likely to be originated from euploid spermatogonia, i.e. the testis shows a mosaic condition where the majority of tubules contains 46,XXY spermatogonia while in a few of them spermatogonia carry a normal chromosomal asset (46,XY) [32].”

Line 135-136: Please also include here that the gonadal development may be affected.

According to the referee suggestion, we have now included that “gonadal development may be affected”.

In general, I find it quite intuitive to have each genetic element associated with NOA to be listed with reproductive consequences, general health, and consequences on offspring’s health. However, I miss a clearer distinction between what is likely to be caused by the testicular phenotype and what might be caused by a phenotype not related to NOA.

We do mention that some comorbidities are related to testosterone deficiency therefore are related to the testicular phenotype. We have now added to the Klinefelter paragraph some examples for testosterone (i.e. testicular phenotype)-related comorbidities. Such a distinction is not necessary for the other genetic factors because a part from azoospermia they do not cause hypotestosteronemia.

Line 206: I find it odd that the authors only touch briefly on other genes that have been proven causal of NOA. For a gene to reach “diagnostic relevance” is quite a somewhat subjective judgement and I suggest that the authors at least mention more genes and pathways potentially involved in NOA. Several recent publications/genes like PMID: 33963445, 34055789, 34347949, and 33980954 could be mentioned.

A part from AR and TEX11 mutation screening there is no other NOA gene which is tested at the routin diagnostic level.  Although we agree that there is a growing number of novel candidate genes, they are not yet included in the diagnostic work-up, hence we are not writing about their clinical consequences. We stated in the introduction that we have very recently published a comprehensive review on genetics of azoospermia (PMID: 33806855) and there are 2 other very recent papers dealing with candidate NOA genes. In the second part we did include novel candidate genes (from available articles published until July 2021) but only if they fulfilled the criteria of “Shared genes between spermatogenesis and tumorigenesis” in humans. We do not consider the 4 papers (the genes included in them) suggested by the referee suitable for this paragraph since mutations in these genes have not been reported in relationship with cancer.